# Circulating Levels of Muscle-Related Metabolites Increase in Response to a Daily Moderately High Dose of a Vitamin D3 Supplement in Women with Vitamin D Insufficiency—Secondary Analysis of a Randomized Placebo-Controlled Trial

**DOI:** 10.3390/nu12051310

**Published:** 2020-05-04

**Authors:** Lise Sofie Bislev, Ulrik Kræmer Sundekilde, Ece Kilic, Trine Kastrup Dalsgaard, Lars Rejnmark, Hanne Christine Bertram

**Affiliations:** 1Department of Endocrinology and Internal Medicine, Aarhus University Hospital, 8200 Aarhus N, Denmark; lars.rejnmark@rm.dk; 2Department of Clinical Medicine, Aarhus University, 8200 Aarhus N, Denmark; 3Department of Food Science, Aarhus University, Agro Food Park 48, 8200 Aarhus N, Denmark; uksundekilde@food.au.dk (U.K.S.); ecekilic90@gmail.com (E.K.); trine.dalsgaard@food.au.dk (T.K.D.); hannec.bertram@food.au.dk (H.C.B.); 4iFOOD, Centre for Innovative Food Research, Aarhus University, 8200 Aarhus N, Denmark

**Keywords:** vitamin D, secondary hyperparathyroidism, skeletal muscle, metabolomics, postmenopausal women

## Abstract

Recently, we demonstrated negative effects of vitamin D supplementation on muscle strength and physical performance in women with vitamin D insufficiency. The underlying mechanism behind these findings remains unknown. In a secondary analysis of the randomized placebo-controlled trial designed to investigate cardiovascular and musculoskeletal health, we employed NMR-based metabolomics to assess the effect of a daily supplement of vitamin D3 (70 µg) or an identically administered placebo, during wintertime. We assessed the serum metabolome of 76 postmenopausal, otherwise healthy, women with vitamin D (25(OH)D) insufficiency (25(OH)D < 50 nmol/L), with mean levels of 25(OH)D of 33 ± 9 nmol/L. Compared to the placebo, vitamin D3 treatment significantly increased the levels of 25(OH)D (−5 vs. 59 nmol/L, respectively, *p* < 0.00001) and 1,25(OH)_2_D (−10 vs. 59 pmol/L, respectively, *p* < 0.00001), whereas parathyroid hormone (PTH) levels were reduced (0.3 vs. −0.7 pmol/L, respectively, *p* < 0.00001). Analysis of the serum metabolome revealed a significant increase of carnitine, choline, and urea and a tendency to increase for trimethylamine-N-oxide (TMAO) and urinary excretion of creatinine, without any effect on renal function. The increase in carnitine, choline, creatinine, and urea negatively correlated with muscle health and physical performance. Combined with previous clinical findings reporting negative effects of vitamin D on muscle strength and physical performance, this secondary analysis suggests a direct detrimental effect on skeletal muscle of moderately high daily doses of vitamin D supplements.

## 1. Introduction

The prevalence of vitamin D insufficiency is high, especially during wintertime [1]. As low levels of 25-hydroxyvitamin D (25(OH)D) are associated with adverse skeletal and non-skeletal health outcomes and as a correction of vitamin D deficiency by supplementation is cheap and feasible, studies elucidating the effects of treating vitamin D insufficiency are obviously of major public health interest.

It is well described that low levels of 25(OH)D may elevate the levels of parathyroid hormone (PTH), causing secondary hyperparathyroidism (SHPT) [2]. High PTH levels are associated with adverse health outcomes independently of low 25(OH)D levels [2,3,4]. Vitamin D insufficiency is pragmatically defined as the level of 25(OH)D below which PTH increases, and SHPT has been suggested as the best marker of vitamin D insufficiency [2,5]. Vitamin D supplementation normalizes the levels of 25(OH)D and PTH, and it has been suggested that individuals with SHPT are more prone to adverse effects of low 25(OH)D levels.

Findings from cross-sectional and cohort studies suggest an inverse association between vitamin D status and adverse health outcomes, whereas data from randomized clinical trials (RCTs) are less conclusive or even report negative effects of vitamin D supplementation on musculoskeletal health [6,7]. The discrepancy between observational studies and RCTs is an indisputable fact [8,9]. So far, many RCTs have been criticized for including participants with a replete vitamin D status, thereby not reflecting the findings from observational studies [9,10].

Metabolomics is a post-genomic advanced method of analysis. Through an explorative approach, metabolomics seeks to characterize and quantify as many metabolites as possible, which constitute the so-called metabolome [11]. The metabolome provides an expression of an individual’s metabolic state, and studies suggest that the metabolome may predict individuals’ different responses to interventions [12].

Taking into account the large number of publications on the role of vitamin D in metabolic health, metabolomics studies are sparse, and RCTs investigating the effect of vitamin D supplements on the metabolome almost non-existing [13].

In the present study, designed to investigate the cardiovascular and musculoskeletal effects of vitamin D supplementation, we applied a nuclear magnetic resonance (NMR)-based approach, to study metabolic changes in otherwise healthy, postmenopausal women with vitamin D insufficiency and relatively high levels of PTH, randomized to a daily oral supplement of cholecalciferol (vitamin D3) of 70 µg (2800 IU) or a similarly administered placebo for 12 weeks during wintertime. Using this explorative method, we hypothesized that a normalization of plasma 25(OH)D associated with a decrease of PTH levels changes the metabolome, providing knowledge of the underlying metabolic pathways involved.

## 2. Materials and Methods

The study was an investigator-initiated parallel group, single-center, randomized double-blinded placebo-controlled trial.

The Danish Data Protection Agency (1-16-02-492-14), the Danish Health Authority (2014-003645-10), the Regional Committee on Biomedical Research Ethics (1-10-72-326-14), and the Danish Health Data Authority (FSEID-00001274) approved the project. The local unit for Good Clinical Practice at Aarhus University Hospital monitored the study. Clinicaltrials.gov: #NCT02572960.

The recruitment of participants has previously been reported in detail [14]. Briefly, a total of 81 healthy postmenopausal women with SHPT (PTH > 6.9 pmol/L) and 25(OH)D levels <50 nmol/L were recruited from the area nearby Aarhus University Hospital, Denmark. Inclusion criteria involved subjects who had not received any treatment with antihypertensives, diuretics, systemic glucocorticoids, nonsteroidal anti-inflammatory drugs, lithium, or anti-osteoporotic drugs. The study was conducted at latitude 56° N during wintertime (between November and April) to prevent cutaneous synthesis of cholecalciferol. Informed consent was obtained from all individual participants included in the study.

The study design is depicted in Figure 1.

The participants received a daily supplement of 70 µg (2800 IU) of cholecalciferol or a similarly administered placebo for 12 weeks. For the first two weeks, the design was 2 × 2 factorial with an angiotensin II receptor blocker (valsartan, 80 mg per day) or similar placebo in order to study the response of PTH to the treatment in the presence of a blockade of the renin–angiotensin–aldosterone system. These findings have previously been reported, showing no impact of the angiotensin II receptor blocker on PTH levels [15].

The overall compliance, as assessed by pill-count, was 99.2%.

As previously reported, the normalization of vitamin D/PTH levels had no effect on most markers of cardiovascular disease (CVD), quality of life, or body composition as assessed by dual energy X ray absorptiometry [14,16], but improved bone microarchitecture and estimated bone strength [17]. Contrary to what expected, the moderately high dose of vitamin D impaired muscle strength (as assessed by hand grip strength and knee flexion) and physical performance (as assessed by the Timed Up and Go test, TUG) [16].

For the present study, samples from five participants (placebo group, *n* = 3, vitamin D group, *n* = 2) were missing, leaving 76 participants for the metabolomics analysis, as shown in Figure 1.

Participants were included based on a biochemical screening indicating plasma 25(OH)D concentration below 50 nmol/L, plasma calcium and creatinine levels below the upper normal limit, and PTH levels above the upper limit of the normal reference range (i.e., >6.9 pmol/L), thereby excluding patients with hypercalcemic primary hyperparathyroidism or other causes of SHPT than vitamin D insufficiency [14].

The samples reported in the study were collected at baseline and at the end of study as fasting blood samples drawn in the morning after an overnight fast.

Metabolomics analyses and analyses of 25(OH)D and 1,25(OH)_2_D were conducted on serum samples, whereas the rest of the analyses were conducted on plasma samples collected in tubes containing lithium heparin. All blood samples were centrifuged at 4000 rpm at 5 °C for 10 min and subsequently stored at −80 °C. To minimize the intra-individual variability, all women rested while lying down for at least 30 min prior to the collection of blood samples.

The total plasma levels of 25(OH)D (25(OH)D2 + 25(OH)D3) and 1,25-dihydroxyvitamin (1,25(OH)_2_D) were quantified using isotope dilution liquid chromatography–tandem mass spectrometry (LC–MS/MS), which is the gold standard for 25(OH)D measurements [18]. Using a second-generation immunoassay on an automated immunoanalyzer (Roche Diagnostics GmbH, Mannheim, Germany), plasma intact PTH was measured in duplicate.

Plasma glucose and lipid profile, as well as measurements of 24 h urine electrolytes were consecutively analyzed using standard laboratory procedures at the Department of Clinical Biochemistry, Aarhus University Hospital Denmark. Participants were verbally informed and received a written instruction from the laboratory prior to urine collection to ensure high quality of the measurements.

Muscle strength was assessed as maximum voluntary isometric muscle strength, with an adjustable dynamometer chair (Good Strength, Metitur Ltd., Jyvaskyla, Finland) [16]. Using a stopwatch, the TUG test provided the time to stand up, walk 3 m as fast as possible in a straight line, and immediately return to the chair [16].

NMR analyses were conducted in October 2019. Prior to the NMR analyses, serum samples were thawed at room temperature, vortexed for 30 s, and filtered using spin filters with a 10 kDa cutoff (13,000× *g* for 90 min at 4 °C; Amicon Ultra-0.5 mL 10K, Merck, Darmstadt, Germany). Prior to use, the filters were washed 3 times with 0.5 mL milliQ H_2_O. A volume of 400 µL of filtrate was transferred to a 5 mm NMR tube (VWR International, Herlev, Denmark) with 100 µL phosphate buffer (pH 7.4, 300 mM) and 100 µL D_2_O containing 0.05 wt. % sodium salt (TSP) (Sigma-Aldrich, Søborg, Denmark). All NMR spectra were acquired at 298 K on a Bruker Avance III 600 spectrometer operating at a ^1^H frequency of 600.13 MHz and equipped with a 5 mm TXI probe (Bruker BioSpin, Rheinstetten, Germany). A one-dimensional (1D) nuclear Overhauser enhancement spectroscopy (NOESY)-presat pulse sequence (noesypr1d) with water suppression was applied, and a total of 64 scans were collected into 32 K data points spanning a spectral width of 12.15 ppm, and relaxation delay was 5 s. Baseline and phase correction of the spectra were done manually using TopSpin 3.0 (Bruker BioSpin). Assignment and quantification of the ^1^H NMR signals was performed by using Chenomx NMR Suite version 8.1 (Chenomx Inc., Edmonton, AB, Canada). The concentration of metabolites was calculated based on the known glucose concentration.

Data distribution was tested using QQ plots and histograms. Normally distributed data were analyzed with parametric tests, and non-normally distributed data with non-parametric tests. Baseline data are reported as means with standard deviation, medians with interquartile (25th, 75th percentiles) range (IQR), or numbers with percentages. The effects of treatment are reported as absolute changes from baseline. Significance was tested using a test for two independent samples or, Mann–Whitney U test, as appropriate. Pearson correlation coefficient (r) was used to assess associations between changes in the serum metabolome and changes in plasma levels of 25(OH)D, as well as changes in fat and lean mass, strength in handgrip, knee flexion at 60°, and the TUG test. We considered a two-tailed *p* value <0.05 as statistically significant. SPSS version 26 was used for the statistical analyses.

## 3. Results

### 3.1. Baseline Characteristics and Effects of Vitamin D Supplementation on 25(OH)D, 1,25(OH)_2_D, and PTH Levels

Baseline characteristic are reported in Table 1. The randomization was well balanced.

Baseline plasma levels of vitamin D, PTH, renal function, and electrolytes, as well as responses to treatment are depicted in Table 2.

At baseline, the plasma levels of 25(OH)D were 33 nmol/L in the total group of women (*n* = 76). At the end of the study, the levels of 25(OH) had increased significantly to 90 nmol/L (95%, CI: 86 to 95) in the vitamin D group as compared with 30 nmol/L (95% CI: 28 to 33) in the control group. Within the vitamin D group, 25(OH)D and 1,25(OH)_2_D increased, and PTH decreased (*p*_all_ < 0.001). Similarly, within the placebo group, 25(OH)D and 1,25(OH)_2_D decreased significantly (*p*_all_ < 0.001), whereas PTH increased (*p* = 0.02, data not shown).

### 3.2. Changes in Muscle-Related Metabolites

The urinary 24 h excretion of creatinine tended to increase in the vitamin D group (0.33 ± 1.53) as compared with the placebo group (−0.37 ± 1.67), *p* = 0.06.

Compared with the placebo, vitamin D supplementation significantly increased the serum levels of carnitine to 6.0 µmol/L (95% CI: −1.1 to 13) vs. −5.5 µmol/L (−13 to 2.0), *p* = 0.03), choline to −0.00 µmol/L (−2.2 to 2.1) vs. −4.1 µmol/L (−6.7 to −1.6), *p* = 0.02), and urea to 45 µmol/L (24 to 66 vs. 13 µmol/L (−7.0 to 34, *p* = 0.03), whereas trimethylamine-N-oxide (TMAO) tended to increase, reaching 6.3 µmol/L (1.5 to 11) vs. 0.6 µmol/L (−2.7 to 4.0, *p* = 0.05), Table 3.

### 3.3. The Effect of Valsartan on the Metabolome

No significant interactions of valsartan with the metabolome were found when the drug was given either alone or in combination with vitamin D (data not shown). In a secondary analysis reporting the effect of valsartan (plus/minus vitamin D) vs. placebo, valsartan (plus/minus vitamin D) did not affect any of the measures of the metabolome (data not shown).

### 3.4. Nutrient Intake and Physical Activity

No differences in estimated daily calcium intake were found (Table 1). There were no differences in the intake of major sources of vitamin D (egg and fish) between the two groups, and no differences in the intake of fruit and vegetables (data not shown). The estimated physically activity did not differ between groups, as previously reported [4].

### 3.5. Correlations between Muscle-Related Metabolites, Body Composision, Musle Strength, and Physical Performance

As reported in Table 4, changes in total fat mass correlated positively with changes in the levels of carnitine (r = 0.29, *p* = 0.01) and urea (r = 0.25, *p* = 0.03). Moreover, changes in the levels of carnitine (r = 0.29, *p* = 0.01), choline (r = 0.23, *p* = 0.04), and urea (r = 0.26, *p* = 0.02) correlated positively with changes in the TUG test, i.e., increases in these metabolites were associated with a longer time spent on performing the test. Changes in handgrip strength were negatively correlated with changes in choline levels (r = −0.25, *p* = 0.05) and excreted creatinine (r = −0.25, *p* = 0.04), i.e., increases in serum choline and excreted creatinine were associated with a decreased handgrip strength.

Bivariate correlation analysis showed no correlation between any of 25(OH)D, choline, carnitine, TMAO, excreted creatinine, and urea and total lean mass/appendicular lean mass index, as assessed by dual-energy X-ray absorptiometry, data not shown.

## 4. Discussion

In this exploratory study, we investigated changes in the human metabolome in response to the normalization of vitamin D levels with a daily moderately high dose supplement of vitamin D during wintertime. Vitamin D supplementation effectively normalized 25(OH)D levels. Compared to placebo, vitamin D supplementation significantly increased the serum levels of carnitine, choline, and urea and tended to increase the serum levels of TMAO and those of creatinine excreted in urine.

Carnitine and choline are two essential nutrients. The major sources of these nutrients are animal products, especially red meat [19,20]. Carnitine is required for energy production, as carnitine acts as a transporter of long-chain fatty acids into the mitochondria to be oxidized and produce energy [19]. Within the body, carnitine is accumulated in the cardiac and skeletal muscles. The content of carnitine in skeletal muscle is about 70-fold higher than in plasma [19]. Supplementation with carnitine is proposed to play a role in muscle health, and supplements are widely used among athletes to enhance performance [19,21]. Choline is required to produce acetylcholine and is used at the neuromuscular junction. Choline deficiency is associated with muscle damage [20]. As with carnitine, skeletal muscle contains a large quantity of choline [22,23].

Choline and carnitine are metabolized by gut microorganisms to produce trimethylamine (TMA), which is subsequently absorbed by the gut and oxidized by flavin-monooxygenases (FMOs) in the liver to produce TMAO [19,20] (Figure 2).

TMAO is mainly known as a waste product of carnitine and choline metabolism [25]. TMAO has received attention as a consequence of a proposed negative effect on cardiovascular health [26], although not all studies support this observation [27]. Intriguingly, the POUNDS Lost trial suggests a positive relationship between circulating TMAO and bone mineralization [28]. Although the 12 weeks of vitamin D supplementation did not affect bone mineral density in our study, we observed improved bone health in terms of increased trabecular thickness and estimated bone strength at the tibia [17].

Blood urea is a product of protein catabolism, and in the urea cycle, nitrogen from the muscles are converted to ammonia and, via liver enzymes, to water-soluble urea, which can be excreted by the kidneys (Figure 2). Creatinine is a breakdown product of creatine from muscle and protein metabolism [29]. We observed that 24 h urinary excretion of creatinine tended (*p* = 0.06) to increase in the vitamin D group.

A possible explanation for the increase in serum carnitine and choline is an increased intestinal absorption. A reduced use and/or degradation is also plausible.

Our findings need to be considered in relation to our clinical findings on cardiovascular and musculoskeletal health [14,16,17]. Overall, there was no effect of vitamin D supplementation on measures of cardiovascular health. Cardiac and/or smooth muscle cells also contain the examined muscle-related metabolites. Thus, we cannot rule out that the metabolites derive from cardiac and/or smooth muscle cells, but in a post hoc analysis no associations were found between the muscle-related metabolites and blood pressure, arterial stiffness, or cardiac conductivity. In contrast, we previously reported a detrimental effect on muscle health and physical performance [16]. Carnitine, choline, TMAO, urea, and creatinine derive from the muscles [29] (Figure 2), and the increase in carnitine, choline, creatinine, and urea correlated negatively with the findings on muscle strength/performance, suggesting that these findings can be ascribed to changes in skeletal muscle.

The amount of carnitine, choline, TMAO, urea, and creatinine are dependent on protein intake, the body’s capacity to catabolize protein, and their adequate excretion by the renal system. There was no between-group difference in renal function, physical activity, or estimated intake of different nutrients [16].

In recent years, an increased number of studies have demonstrated adverse effects of higher dosages of vitamin D. Negative effects are mainly reported on muscle strength and risk of falls [16,30,31,32,33,34,35,36]. A recent study with vitamin D3, 70 µg per day, suggested a negative effect on lean body mass [30].

Previous studies have reported an increased risk of falls in response to vitamin D supplementation [31,32,33,36]. Orthostatic hypotension due to decreased activity of the renin–angiotensin–aldosterone system has been suggested, but in this study, markers of this system were not affected by vitamin D supplementation [14].

To the best of our knowledge, higher dosages of vitamin D supplementation has not been reported to impair postural stability [16,30,37,38], and it therefore seems most likely that the increased risk of falls is attributable to an impaired muscle strength/function.

The mechanisms behind the studies reporting negative effects of vitamin D have not yet been fully elucidated. Vitamin D receptors are almost ubiquitously expressed in human tissues. Over-expression of vitamin D receptors and inadequate differentiation of muscle fibers are reported in response to active vitamin D in supra-physiological dosages [39,40]. Elevated levels of creatine kinase (which converts creatine to creatinine) [40] as well as fat infiltration are also reported [41].

Together with the existing data, this explorative study suggests a direct detrimental effect on skeletal muscles causing a leak of muscle products to the blood stream and subsequently to urine (creatinine).

That vitamin D could heal myopathies was a clinical observation before it was possible to measure 25(OH)D levels, and threshold levels are largely based on findings from observational studies. In general, randomized clinical trials have largely failed to demonstrate any effect of vitamin D. Possibly, vitamin D has divergent effects on different tissues (e.g., an increased risk of falls and thereby increased risk of fractures despite an improved bone health), which overall counterbalance each other.

The present study has several strengths as well as limitations. Most importantly, the well-balanced randomized placebo-controlled design conducted during wintertime in women with low levels of 25(OH)D and relatively high PTH levels leaves a unique study group not previously investigated with respect to NMR-based metabolomics. The NMR-metabolomics data were not a pre-planned endpoint, and there was no a priori hypothesis. As the study is exploratory, it is important to establish that the study is hypothesis-generating rather than hypothesis-testing. We did not adjust for multiple testing as this is a rather conservative approach, which lowers the chance of detecting potential associations. This is, on the other hand, a major limitation, and we cannot rule out that some of the findings are type I errors.

The three months duration of the intervention is relatively short.

The initial factorial design is a limitation. There was no interaction between vitamin D and valsartan in the reported outcomes, and valsartan did not affect any of the measures of the metabolome, but as the half-life of valsartan is 6–9 h, a potential effect of valsartan on the metabolome in the reported 12-week measures is unlikely [42].

The estimated intake of vitamin D was not calculated, and neither was estimated intake of meat. Finally, PTH level at baseline was substantial lower than expected. This has previously been discussed in details [14].

Unfortunately, we did not use different dosages of vitamin D or assess muscle health with biopsies.

The dose used to treat vitamin D insufficiency was larger than recommended in most guidelines, and our results do not allow for conclusions on lower dosages of vitamin D [19]. Detrimental effects on muscle health are not reported with dosages below 20 µg/day [16]. In contrast, dosages at 70 µg/day are commonly used to treat insufficiency.

At present, it is unknown whether it is the rapid increase in 25(OH)D or the levels of 25(OH)D at the end of the study that cause potential adverse effects on skeletal muscle and/or falls. “Very high dose bolus studies” [33,36] has been reproduced by studies using moderately high daily dosages of vitamin D3 [16,30,32], also in participants with vitamin D insufficiency [16,32]. The fact that adverse effects have been reported also for levels of 25(OH)D within the reference range suggests that a rapid increase in 25(OH)D is associated with adverse effects on skeletal muscles and/or falls. The mechanism behinds those findings needs to be established.

In 2011, the upper tolerance limit was increased from 50 to 100 µg/day based on the lack of occurrence of hypercalcemia. Data on falls and muscle health from 2010 on (mainly from 2015) suggest a reduction of the upper tolerance limit [16,30,31,32,33,34,35,36].

## 5. Conclusions

Normalization of 25(OH)D levels with a moderately high daily dose of vitamin D supplementation during wintertime causes changes in the metabolome in terms of increased serum levels of carnitine, choline, and urea and a tendency towards increased serum levels of TMAO and urinary creatinine. This study suggests a potential detrimental effect of vitamin D supplements on skeletal muscles, with leak of muscle products to the circulation.

## Figures and Tables

**Figure 1 nutrients-12-01310-f001:**
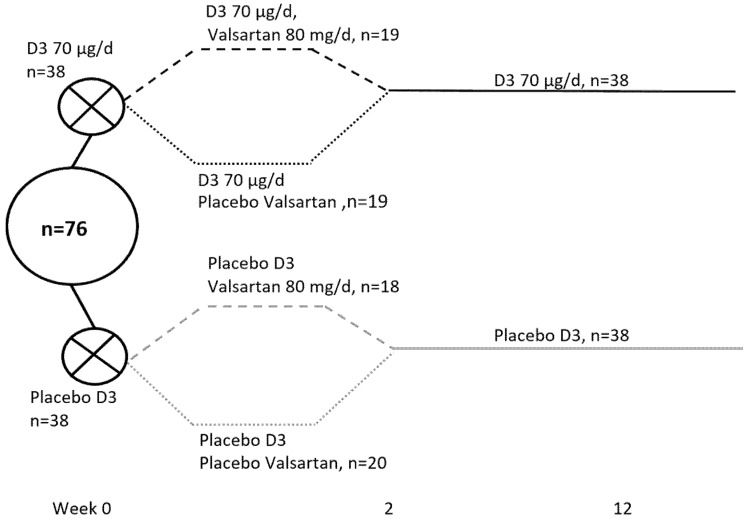
Study design: All women were included from November to February to avoid cutaneous vitamin D synthesis (latitude 56° N).

**Figure 2 nutrients-12-01310-f002:**
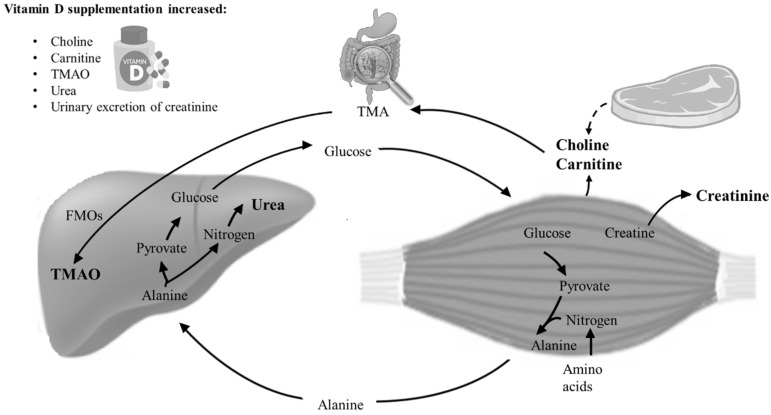
Schematic illustration of the metabolic pathways of the significant and border-significant findings on choline, carnitine, creatinine, TMAO, and urea. Choline and carnitine are nutrients normally ingested through protein-rich diets. TMAO is generated from the hepatic oxidation of trimethylamine (TMA), formed by the gut microbiota from carnitine and choline. In the body, high concentrations of choline and carnitine are present in skeletal muscle. In the glucose–alanine cycle, amino groups and carbons from skeletal muscle are transported to the liver. In the liver, alanine is converted to pyruvate and nitrogen. Nitrogen enters the urea cycle, and pyruvate is used to produce glucose [24]. Creatinine is a waste product of a non-enzymatic degradation of creatine phosphate, serving as a reserve of high-energy phosphates in skeletal muscle. Together with previous clinical findings on muscle strength and physical performance, the data suggest that the increase in choline, carnitine, creatinine, TMAO, and urea, all waste products originating from muscle catabolism, may be caused by a direct toxic effect on skeletal muscle.

**Table 1 nutrients-12-01310-t001:** Baseline characteristics. Data are reported as mean ± SD, median with interquartile (25th, 75th percentiles) range (IQR) or numbers (%). HDL, high-density lipoprotein, LDL, low-density lipoprotein.

	Vitamin D (*n* = 38)	Placebo (*n* = 38)	*p*-Value
**Age and body composition**			
Age (years)	64.5 [61.0; 68.25]	65.5 [62.0; 68.25]	0.56
Body weight (Kg)	75.3 [67.3; 90.3]	70.4 [65.0; 78.2]	0.17
Height (cm)	166.2 ± 4.7	165.1 ± 6.0	0.39
Body mass index (Kg/m^2^)	27.3 [23.3; 32.0]	26.8 [23.6; 28.8]	0.42
Appendicular lean mass index (Kg/m^2^)	10.8 [10.0; 12.1]	10.7 [10.1; 11.5]	0.56
Fat mass index (Kg/m^2^)	18.7 [14.1; 23.8]	17.9 [12.5; 20.2]	0.16
**Indices of bone health**			
Calcium intake (mg/day)	850 [700; 950]	700 [650; 1075]	0.87
History of fracture in adulthood *n* (%)	13 (34)	7 (20)	0.12
**Smoking status** ***n*** **(%)**			0.22
Never	23 (61)	22 (68)	
Current	1 (3)	5 (13)	
Former	14 (37)	11 (29)	
**Use of medication**			
Any *n* (%)	13 (34)	13 (34)	0.60
**Indices of cardiovascular health**			
Systolic 24 h blood pressure (mmHg)	129 [125; 146]	128 [118; 135]	0.14
Diastolic 24 h blood pressure (mmHg)	75 [68; 83]	74 [68; 79]	0.31
Total cholesterol (mmol/L)	5.2 [4.8; 5.9]	5.5 [5.0; 6.4]	0.12
HDL (mmol/L)	1.8 ± 0.4	1.8 ± 0.4	0.76
LDL (mmol/L)	3.0 ± 0.8	3.3 ± 1.0	0.14
Triglycerides (mmol/L)	1.1 [0.7; 1.3]	0.9 [0.7; 1.3]	0.65
Arterial stiffness (m^2^/s)	9.7 ± 1.7	9.1 ± 1.3	0.10

**Table 2 nutrients-12-01310-t002:** Baseline levels of 25(OH)D, 1,25(OH)_2_D, PTH, and electrolytes. Baseline data are reported as mean ± SD or median with IQR (25%–75% percentile). The mean of the entire group is reported at baseline, as there was no difference between the groups in any of the measurements. Changes are reported as means ± SD. Significant results are shown in bold.

			Changes (Δ)		
	Ref.Range	Baseline, *n* = 76	Vitamin D, *n* = 38	Placebo, *n* = 38	*p*-Value
Plasma					
25(OH)D (nmol/L)	50–160	33 ± 9	58.5 ± 16.3	−4.5 ± 6.3	**<0.00001**
1.25(OH)2D (pmol/L)	60–180	53 ± 14	18.5 ± 15.2	−9.6 ± 9.9	**<0.00001**
PTH (pmol/L)	1.6–6.9	6.1 ± 1.3	-0.69 ± 0.79	0.28 ± 0.80	**<0.00001**
Ca^2+^ (mmol/L)	1.18–1.32	1.25 ± 0.04	0.00 ± 0.04	−0.01 ± 0.03	0.20
Magnesium (mmol/L)	0.7–1.1	0.88 ± 0.06	-0.01 ± 0.04	0.01 ± 0.04	0.24
Phosphate (mmol/L)	0.76–1.41	1.00 ± 0.14	0.06 ± 0.11	0.04 ± 0.12	0.52
eGRF	>60 mL/min	82.4 [73.1; 90.7]	−2.18 [−5.45; 4.21]	−1.15 [−5.30; 1.76]	0.94
Urine					
Creatinine (mmol/24 h)	6–15	10.3 ± 1.9	0.33 ± 1.53	−0.37 ± 1.67	0.06

Abbreviations: 25(OH)D, 25-hydroxy vitamin D, 1,25(OH)_2_D, 1,25dihydroxy vitamin D, PTH, parathyroid hormone, Ca^2+^, ionized calcium, eGFR, estimated glomerular filtration rate. Significant results are shown in bold.

**Table 3 nutrients-12-01310-t003:** Significance of the changes in metabolites level observed in fasting serum after a 12-week intervention with vitamin D supplementation (70 µg/d) compared to placebo.

Metabolites	Baseline, µmol/L, *n* = 76	Changes (Δ), µmol/L		*p*-Value
		Vitamin D, *n* = 38	Placebo *n* = 38	
Hydroxybutyrate	38 [22; 90]	−0.34 [−55; 25]	2.1 [−34; 29]	0.58
Acetate	16 [9; 23]	−2.0 [−11; 2.8]	−0.33 [−12; 2.7]	0.92
Acetoacetate	18 [10; 29]	1.4 [−14; 14]	2.4 [−7.8; 10]	0.93
Acetone	7.1 [5.6; 11]	−1.3 [−7.4; 1.3]	−1.0 [−3.1; 1.8]	0.33
Alanine	210 ± 57	8.3 [−15; 47]	7.1 [−30; 38]	0.26
Betaine	16 [9; 23]	3.3 (−0.9 to 5.5)	0.3 (−2.7 to 3.3)	0.36
Carnitine	77 [65; 86]	6.0 (−1.1 to 13)	−5.5 (−13 to 2.0)	**0.03**
Choline	15 ± 6	−0.00 (−2.2 to 2.1)	−4.1 (−6.7 to −1.6)	**0.02**
Citrate	93 [82; 108]	−0.42 (−8.9 to 8.0)	−1.1 (−7.8 to 5.7)	0.91
Creatine	23 [18; 31]	3.5 (−0.44 to 7.5)	4.2 (−0.76 to 9.1)	0.84
Creatinine	49 ± 11	8.6 (3.4 to 14)	5.3 (1.7 to 8.9)	0.30
Dimethylamine	1.5 [1.0; 4.3]	0.3 [0.0; 0.8]	0.1 [−0.30; 0.5]	0.25
Formate	1.3 [0.8; 2.1]	0.0 (−0.24 to 0.3)	−0.01 (−0.29 to 0.3)	0.91
Glucose	5500 [5125; 6000]	0.0 [−225; 300]	−100 [−325; 100]	0.31
Glutamate	48 [35; 73]	−2.1 [−23; 18]	−14 [−28; 7.0]	0.15
Glutamine	470 [410; 520]	64 [14; 130]	60 [15; 127]	0.82
Glycerol	390 [320; 530]	0.0 [−180; 80]	−34 [−160; 71]	0.65
Glycine	69 ± 22	6.4 (−0.30 to 13)	2.9 (−5.7 to 11)	0.52
Isoleucine	67 [59; 80]	4.6 [−6.1; 19]	4.4 [−10; 12]	0.19
Lactate	750 [650; 970]	77 (−31 to 180)	10.7 (−92 to 110)	0.37
Leucine	150 [140; 170]	12 [−8.2; 36]	−2.5 [−20; 24]	0.17
Lysine	260 [240; 280]	13 [−30; 60]	6.2 [−8.3; 46]	0.83
Methionine	19 [17; 26]	−1.8 [−8.6; 2.1]	−2.7 [−7.6; 0.8]	0.58
OPhosphocholine	19 [16; 24]	−1.2 (−2.7 to 0.4)	−1.2 (−2.9 to 0.5)	0.98
Ornithine	83 [61; 110]	5.9 (−11 to 23)	5.0 (−8.0 to 18)	0.93
Phenylalanine	57 [51; 65]	4.7 (−0.1 to 9.6)	0.4 (−3.5 to 4.4)	0.17
Proline	150 [110; 280]	52 (27 to 77)	25 (2 to 47)	0.11
Pyruvate	12 [7; 18]	4.5 (1.7 to 7.4)	4.9 (1.9 to 7.9)	0.87
Succinate	3.6 [2.6; 6.6]	−0.25 [−2.9; 0.6]	−0.9 [−4.7; 1.4]	0.87
Threonine	83 [74; 91]	8.2 [−8.3; 17]	6.8 [−2.6; 18]	0.89
TMAO	36 [31; 42]	6.3 (1.5 to 11)	0.6 (−2.7 to 4.0)	**0.05**
Tyrosine	59 ± 15	3.0 [−5.5; 12]	1.0 [−4.2; 9.0]	0.32
Urea	180 [150; 220]	45 (24 to 66)	13 (−7.0 to 34)	**0.03**
Valine	240 [200; 280]	14 [−5.0; 46]	0.9 [−21; 31]	0.11
τMethylhistidine	110 [100; 120]	3.6 [−13; 30]	4.8 [−13; 30]	0.34

The metabolites were quantified by ^1^H NMR spectroscopy. Except from choline (vitamin D, 14 ± 16 vs. placebo 17 ± 6.2, *p* = 0.02), none of the data at baseline differed between groups when stratified by treatment allocation. The mean ± standard deviation or median (25th, 75th percentiles) for the whole group is reported. Changes were calculated as individual post-intervention values minus baseline values for each metabolite, and data are reported as median (25, 75 percentiles) or mean with 95% confidence intervals. Abbreviation: TMAO, trimethylamine N-oxide. Significant results are shown in bold.

**Table 4 nutrients-12-01310-t004:** Correlations between changes in the levels of 25-hydroxyvitamin D, carnitine, choline, and urea, as well as 24 h renal excretion of creatinine and previoulys reported significant markers of muscle health [16] and body composition (*n* = 76). A positive correlation at the TUG test means spending longer time performing the test (worse performance).

Changes (Δ)	Total Fat Mass		TUG		Handgrip Strength		Knee Flexion 60°	
	r	*p*-Value	r	*p*-Value	r	*p*-Value	R	*p*-Value
25(OH)D, nmol/L	-	-	-	-	−0.27	0.03	−0.29	0.02
Carnitine, mmol/L	0.29	0.01	0.29	0.01	-	-	-	-
Choline, mmol/L	-	-	0.23	0.04	−0.25	0.04	-	-
Urea, mmol/L	0.25	0.03	0.26	0.02	-	-	-	-
Urine creatinine, mmol/day	-	-	-	-	−0.26	0.04	-	-

Abbreviations: r: Pearson correlation coefficient, 25(OH)D: 25-hydroxyvitamin D, TUG: Time Up and Go test, knee flexion 60°: maximum voluntary muscle strength with the knee flexed 60° from the fully extended leg.

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
