# Peer review of "Circulating Levels of Muscle-Related Metabolites Increase in Response to a Daily Moderately High Dose of a Vitamin D3 Supplement in Women with Vitamin D Insufficiency—Secondary Analysis of a Randomized Placebo-Controlled Trial"

_nutrients, 2020, doi:10.3390/nu12051310_

Round 1

Reviewer 1 Report

This is a carefully conducted randomized controlled trial which provides important secondary data possibly contributing to an understanding of the adverse effects of high doses of vitamin D on muscle. I have only minor comments as follows:

  1. For the metabolomics data, baseline values should be shown as well as changes so that the percent change can be deduced.
  2. I cannot see a description of the statistical methods. What adjustment was made for the large number of variables assessed in the metabolomics analysis.
  3. In the Abstract “pall<0.0001” Requires revision to make its meaning clear.
  4. There are a small number of typographical and grammatical errors. The manuscript requires careful proofreading.

Reviewer 2 Report

Bislev et al. present interesting metabolomic muscle-related data in healthy postmenopausal women out of a randomized controlled study von vitamin D and/or valsartan and placebo. The study has been published for the main parts in 2018.

The manuscript is well written and contains important information. However, there are important points to clarify.

Major comments

  • Importantly, the design of the study (Figure 1) shows 38 persons in each arm of the study, where 19 (20 respectively) are on either vitamin D supplementation alone or on placebo (PBO). As all others (!) are getting valsartan as of the Clintrial definition, it is not allowed to include them in the overall analysis and even more not mentioning this anyhow, as this medication might be involved in any of the cited pathways. Therefore, all analyses have to be performed for the vitamin D vs. PBO group ONLY. Do the differences in the results stay the same?
  • When interpreting differences in muscle metabolism, the authors reflect on skeletal muscle metabolism only – which excludes heart and other muscle types. How would they be able to dissect skeletal muscle from heart (or other) muscle involvement, since these subjects might have other involvements as well?
  • The term “high dose” vitamin D is irritating, since this evokes a misleading interpretation by the readers due to very high-dose studies in the past. This regimen is routinely used by many colleagues and should not be called “high dose” – the patient levels after vitamin D supplementation show no extreme increases whatsoever. The discussion points have to be put into perspective in view of REALLY high dose vitamin D features.
  • What was the intention to use serum for the NMR analyses instead of plasma?
  • Apart from the investigative character of this study, the authors may open some explanation models based on recent publications, why their metabolomic associations might occur. A “toxic effect” of normal vitamin D levels are not credible.

Minor comments

  • There are some misprintings, e.g. line 8, 18, 23, 70 etc. Please read through the manuscript after the revision
  • The discussion contains a lot of information (e.g. line 198-223) which should be part of the introduction. Please change accordingly.

Reviewer 3 Report

The authors claim that they had demonstrated negative effects of vitamin D supplementation on muscle strength and physical performance in women with secondary hyperparathyroidism (SHPT). This manuscript is a secondary analysis of the randomized placebo controlled trial designed to investigate cardiovascular- and musculoskeletal health, they employed NMR-based metabolomics to assess the effect of a daily supplement of vitamin D3, 70μg or identical placebo, during wintertime. 76 postmenopausal women with SHPT due to vitamin D insufficiency were studied. Analysis of the serum metabolome revealed a significant increase in carnitine, choline, and urea and tended to increase trimethylamine-N-oxide (TMAO) and urinary excretion of creatinine. This suggests a direct detrimental effect on skeletal muscle of high dose vitamin D supplements. The study design is inadequate, but the rationale is sound. There are some concerns:

  1. From the entire screened cohort, the authors only included participants women with 25(OH)D level below < 50 nmol/l, a PTH level above upper limit of the reference range ( > 6.9 pmol/l) as well as plasma levels of creatinine and calcium below upper reference range. How did the authors exclude the true primary hyperparathyroidism patients who may also show low vit-D levels? Did they receive parathyroid sonography or MIBI scan?
  2. The citation of reference 14 was not completed. It should be: Endocrine. 2018 Oct;62(1):182-194.
  3. Previous studies have reported an increased risk of falls in response to vitamin D supplementation. These risks of falls may related to decrease activity of the RAA system which may contribute to posture hypotension in elderly patients.
  4. Figure 2 illustration of the metabolic pathways of the significant of border-significant findings of choline, carnitine, creatinine, TMAO and urea. The role of vit-D supplement should be added.

Reviewer 4 Report

Bislev and Co-Workers present data of a post-hoc analysis (evaluating the  a priori not planned NMR-measured metabolome) of a randomized placebo controlled trial. 76 postmenopausal women with vitamin D insufficiency were treated either by cholecalciferol or plazebo for 12 weeks. At baseline and after 12 weeeks various metabolites were measured using 1H NMR spectroscopy. The authors found that normalization of 25(OH)D levels with high dose oral supplementation changed the metabolome with increased Serum levels of carnitine, choline and urea. Based on these findings it was concluded that the study results suggest a potential detrimental effect of high dose oral cholecalciferol supplementation (or the 25(OH)D normalisation) on the skeletal muscle which could explain earlier findings on muscle strength published by the same group.

I have the following major comments:

1.) From the data presented (Table 2) and their earlier publication (ref. #16) definitely more than 50% of women were NOT hyperparathyroid at baseline. I know the authors have already discussed this point, but throughout the manuscript the authors  stress that the women had SHPT due to vitamin D insufficiency. I believe it is sufficient to characterise the study cohort as postmenopausal women with vitamin D insufficiency.

2.) In the Methods section I am missing the description of statistical methods used.

3.) Which statistical tests were used to compare differences between Treatment groups and differences in chnages between groups (concerning results presented in Tables 2 and 3)? For PTH median (IQR) should be presented, and I assume a non-parametric test was used (due to variation and low sample size).

4.) The authors compare the differences (between the 2 groups) in changes of metabolites between BL and week 12. They did this for 35 metabolites but (as discussed by the authors) did not adjust for multiple testing. But by random chance the authors might get 1 to 2 "statistically significant" results with α=5% (1 significant result with 20 tests). As the authors use statistical tests to strengthen their argument and claim significant changes they should either use them correctly (with correction for multiple testing) or without and descriptive only.

5.) Kidney function plays a key role in the metabolism and excretion for example of carnitine, urea and creatinine influencing their palsma levels. What was kidney function (GFR) at baseline and after 12 weeks? Did GFR change?  The authors state in the discussion that "there was no between group difference in renal function" citing their earlier work (ref. #16) but I cannot find renal data in the cited paper neither. There it is only mentioned that "women were offered a blood screening to clarify whether …. Plasma levels of …. creatinie was below upper refence levels". Changes in kidney function and highly variable dietray intake could influence minor changes over 12 weeks. With GFR data this could be adjusted for. Because Metabolite level changes were calculated as individual post intervention minus baseline values, individual GFR changes might influence the results even if avergae GFR did not change between both groups over the 12 weeks.

Minor remarks:

1.) I suggest to use µmol/l instead of mmol/l with 3 decimal places in Table 3 (as the metabolites are in the µmol/l reference range).

2.) Is a mean 0.000 mmol/l increase (as for choline) of relevance and really describing a vitamin D-induced plasma level increase? Or did vitamin D supplementation significantly PREVENT choline decrease (as found in the placebo group), which would change the conclusion?

3.) References # 33 and 37 are the same.

Round 2

Reviewer 2 Report

Balanced rebuttal.

Author Response

Thank you

Reviewer 4 Report

The statistical methods are now included.  Unpaired t-test (two independent samples test) for obviously normally distributed differences and Wilcoxon test as non-parametric test were used. The Wilcoxon test (rank sum , signed rank) is a non-parametric test for the comparison of related or matched samples or for reapeated measurements. But as I understand (and as shown in Tables 2 and 3) the mean differences between the two unrelated groups were compaired.

The GLM with models including vit D and valsartan interaction make sense for the authors´ earlier publication, but nor for this post hoc analysis

Concerning my 3rd remark, the authors refer to their answer to Question 2 of Reviewer 1.  But again I am very concerned about doing 35 single group comparisons (using single unpaired t-tests or non-parametric equivalents) in 2 groups of only 38 patients each without correction for these multiple testings and finding 3 statistically significant (p<0.05) results. With one of them (choline; - 0.003 µmol/l in the Vitamin D group) showing no INCREASE by vitamin D compared to baseline. So in my opinion, the conclusions are based on weak evidence.

The results presented in Table 2 are not new (included in their earlier publication of this study), so the main results are the findings of Table 3. But those are methodologically questionable..

The authors might have misunderstood my suggestion for Table 3: I suggested to use µmol/L (with 2 decimal places sufficient) INSTEAD of mmol/L with 3 decimal places.

Minor remark:

Hydroxybutyrate in Vitamin D group: -0.003 mmol/L are -3.40 µmol/l not -0.340; the IQR is also incorrect
